# Adapting to time: Why nature may have evolved a diverse set of neurons

**Karim G. Habashy** [1] *, **Benjamin D. Evans** [2], **Dan F. M. Goodman** [3], **Jeffrey S. Bowers** [1] *

**1** School of Psychological Science, University of Bristol, Bristol, South West England, United Kingdom,
**2** Department of Informatics, School of Engineering and Informatics, University of Sussex, Brighton, East Sussex, United Kingdom, **3** Department of Electrical and Electronic Engineering, Imperial College London, London, London, United Kingdom

* karim.g.habashy@gmail.com (KGH); j.bowers@bristol.ac.uk (JSB)

**Data Availability Statement:** The data in this manuscript is simulated using the code stored in the referenced GitHub repository, and can be generated by running the simulation code.

## Abstract

Brains have evolved diverse neurons with varying morphologies and dynamics that impact temporal information processing. In contrast, most neural network models use homogeneous units that vary only in spatial parameters (weights and biases). To explore the importance of temporal parameters, we trained spiking neural networks on tasks with varying temporal complexity, holding different parameter subsets constant. We found that adapting conduction delays is crucial for solving all test conditions under tight resource constraints. Remarkably, these tasks can be solved using only temporal parameters (delays and time constants) with constant weights. In more complex spatio-temporal tasks, an adaptable bursting parameter was essential. Overall, allowing adaptation of both temporal and spatial parameters enhances network robustness to noise, a vital feature for biological brains and neuromorphic computing systems. Our findings suggest that rich and adaptable dynamics may be the key for solving temporally structured tasks efficiently in evolving organisms, which would help explain the diverse physiological properties of biological neurons.

## Author summary

The impressive successes of artificial neural networks (ANNs) in solving a range of challenging artificial intelligence tasks have led many researchers to explore the similarities between ANNs and brains. One obvious difference is that ANNs ignore many salient features of biology, such as the fact that neurons spike and that the timing of spikes plays a role in computation and learning. Here we explore the importance of adapting temporal parameters in spiking neural networks in an evolutionary context. We observed that adapting weights by themselves (as typical in ANNs) was not sufficient to solve a range of tasks in our small networks, and that adapting temporal parameters was needed (such as adapting the time constants and delays of units). Indeed, we showed that adapting weights is not even needed to achieve solutions to many problems. Our findings may provide some insights into why evolution produced a wide range of neuron types that vary in terms of their processing of time and suggest that adaptive time parameters will play an important role in developing models of the human brain. In addition, we showed that

Accessible at: https://github.com/biocomplab/
Neuro-morphology.

**Funding:** This project has received funding from
the Engineering and Physical Sciences Research
Council (EPSRC) New Horizons call (Reference EP/
X017915/1) to KGH and the European Research
Council (ERC) under the European Union's Horizon
2020 research and innovation programme (grant
agreement no. 741134). The funders had no role in
study design, data collection and analysis, decision
to publish, or preparation of the manuscript.

**Competing interests:** The authors have declared
that no competing interests exist.

adapting temporal parameters makes networks more robust to noise, a feature that can
prove beneficial for neuromorphic system design.

## Introduction

Neurons spike, and the timing of spikes matters in neural computations [1–5]. However, it is
notable that most computational models of neural systems ignore spikes and spike timing,
relying on the finding that the first principal component of neural information is in the firing
rate [6]. For example, Artificial Neural Networks, such as convolutional networks and trans-
formers, use rate coding, with units taking on real valued activation levels that discard all tem-
poral information. As a consequence, researchers using ANNs as models of brains assume that
rate coding is a good abstraction that can support neural-like computations across a variety of
domains, including vision, language, memory, navigation, motor control, etc. [7–10]. Further-
more, because there are no spikes, most ANN learning algorithms ignore the role of fine tem-
poral information and are restricted to changing spatial parameters in the network, namely
weights and biases. Similarly, networks that are built through evolutionary algorithms gener-
ally ignore spike timing, assume rate coding, and typically only adapt the weights and biases.
Even most spiking neural networks (SNNs) only learn through the adaptation of weights and
biases. That is, in most cases, models do not allow for learning and adaptation to extend to the
dimensions of time, such as modifying the conduction delays and time constants of neurons
(although see for example [11–13]).

However, there is strong evidence that adaptive processes modify the neural processing of
time. For example, there is growing evidence for myelin plasticity, in which conduction times
of neurons are modified in adaptive ways to support motor control [14], the preservation of
remote memories [15], spatial memory formation [16], and more generally, myelination is
argued to be a plastic process that shapes learning and human behavior [17]. In addition to
these examples of in-life learning, and more relevant to the current project, evolution has pro-
duced neurons that vary dramatically in their morphology in ways that impact their processing
of time, partly due to the diverse set of axono-dendritic structures [18, 19]. For example, con-
duction rates of neurons vary by over an order of magnitude [20] and time constants of neu-
rons by almost two orders of magnitude [21, 22]. That is, evolution appears to have produced a
diverse set of neuron types to exploit the dimension of time. An archetypal example of this is
the method by which barn owls perform sound localisation [23]. By contrast, with few excep-
tions, units in artificial neural networks are identical to one another apart from their connec-
tion weights and biases.

There is some work with spiking neural networks that explores the adaptive value of
learning time-based parameters. For example, it has been shown that adapting time con-
stants in addition to weights improves performance on tasks with rich temporal structure
[12]. Interestingly, the learned time constants showed distributions that approximate those
observed experimentally. With regard to delays, the traditional approach has been to couple
non-learnable (fixed) delays with spike-timing dependent plasticity (STDP) and study their
combined computational properties [24, 25]. More recently, researchers have developed
methods of learning delays [13, 26, 27]. However, this research has only adapted a single
temporal parameter along with weights, and as far as we are aware, no one has adapted
delays through evolutionary algorithms. Accordingly, it remains unclear how these different
temporal mechanisms interact and we do not have a clear picture of their computational

advantages in an evolutionary context that might help explain the physiological diversity of neurons.

Here we ask whether adapting the temporal parameters of units (axonal delays, synaptic time constants, and bursting), in addition to a spatial parameter (synaptic weights), in spiking neural networks using a simple evolutionary algorithm improves model performance on a range of logic problems that involve a diverse set of input-output mappings. We do this by training networks on a series of tasks of increasing temporal complexity, from semi-temporal binary logic problems (mapping spike trains to spike counts) to fully spatio-temporal tasks (mapping spike trains to spike trains). We find that: (1) in these temporally structured tasks, trainable temporal mechanisms are essential to be able to perform the tasks; (2) there are significant advantages in terms of performance and training robustness to co-evolving multiple mechanisms; (3) adaptive temporal mechanisms provide robustness to noise in both inputs and parameters. These findings provide a proof of principle for the advantages of adapting temporal parameters in natural evolution, may help explain why spatio-temporal heterogeneity is so widespread in nervous systems, and may inform the design of efficient neuromorphic hardware.

## Methods

First, we describe the network architecture and input-output encodings used across simulations, followed by a description of the neuronal model, and finally the evolutionary algorithm we employed.

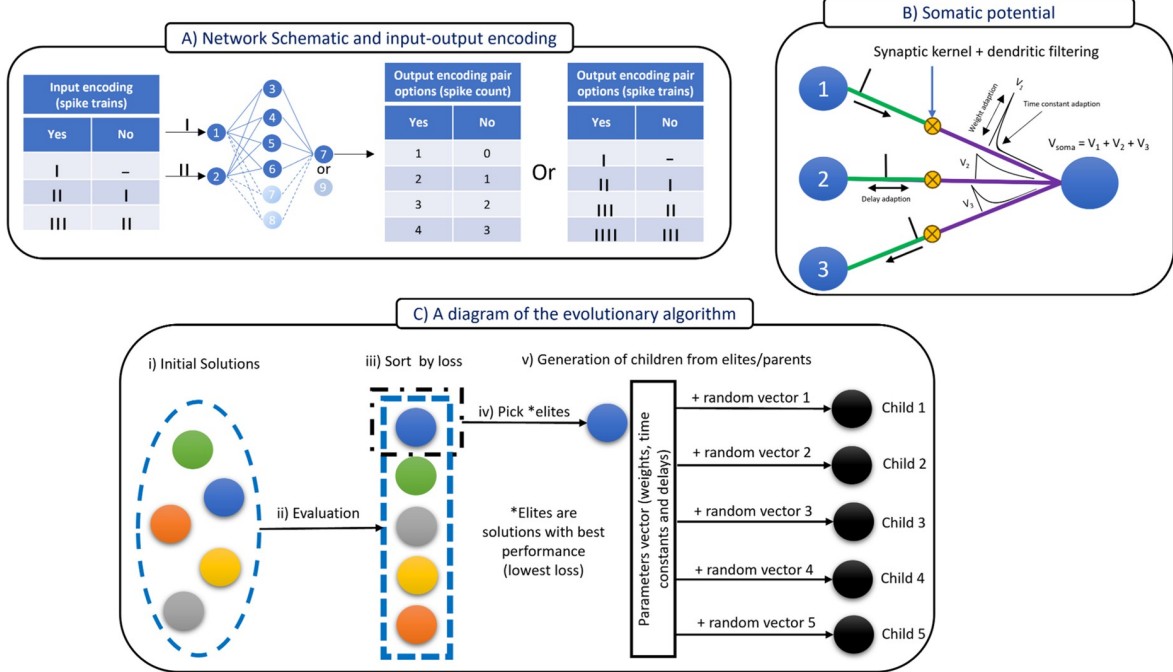

**Fig 1. Schematic of the framework for this study.** (A) The network architecture and the input-output encoding pair. (B) The neuron model illustrating the adaptable parameters and computation of the somatic voltage. (C) Steps of the evolutionary algorithm, which includes: initialization, evaluation, sorting by loss, selection of elites and the generation of a new child population.

## The network architecture and input-output encoding

For all simulations we used a feedforward network with two input neurons, four to six hidden neurons and one output neuron, as illustrated in Fig 1A. We chose to train models on logic problems (XOR, XNOR, OR, NOR, AND and NAND) in small networks as they offer a tractable context for investigating the impact of various neuronal parameters and their role in neural computation across a wide range input-output mappings, including nonlinearly-separable output mappings.

We varied input-output encodings across simulations. The inputs constituted spike trains containing zero to five spikes. This type of encoding can be thought of as burst coding, where we feed the input layer a single burst and test the impulse response of the network. Technically, a real input spike train can be considered as a sequence of bursts, thus this formalism assumes that the timing between bursts is large enough to warrant independent analysis. In this work, this singular burst is padded with silent periods of up to 20 ms on both sides of the burst. Meanwhile, the output code can be a spike count (Fig 1A left output table) or a spike train of a singular burst (Fig 1A right output table). In a spike count, the timing of the spikes does not matter. When the input-output encodings are both spike trains, we can think of the network as engaging in spatio-temporal mapping [28, 29].

## The neuronal model

The building block of the evolved networks was a modified version of the leaky integrate-and-fire neuron (LIF). We introduced three modifications: i) axonal conduction delays were either increased or decreased. This effect is manifested through a synaptic parameter $D_i^L$, where $D$ stands for the total delay, $L$ is the layer index and $i$ is the neuron index. The effective axonal delay is the sum of a default delay (the same for all neurons) and an adaptable 'change in delay' parameter which can take positive or negative values. A positive change in delay is analogous to a slow down in the transmission speed. This is manifested as a shift to the right in the spiking plots shown later and vice versa for a negative change. In brief, we mutate the change in delay and add it to the default value to obtain the effective delay. Technically, the total delay that the soma experiences is the aggregate effect of the pre-synaptic axon and the dendrite, however we simply combine these two and describe them collectively as axonal delays. ii) The leak/decay was shifted from the soma to the dendrites/synapses. This formulation is not new, but employed before in [30, 31]. Technically, there are two main sources of decay in real neurons, one in the dendrites and the other in the soma. Since the dendritic decays are slower due to the low pass filtering induced by the whole associated axon-dendritic structure ([32] Chapter 4), for simplicity, tractability and speed, we only include the dendritic decay and treat the somatic decay as effectively instantaneous relative to the dendritic one. Note, synaptic and somatic time constants converge when all the synapses have the same time constant. iii) In some simulations we included a spike Afterpotential (AP) that is added to the total somatic voltage after a neuron fires [33]. The role of this AP voltage is to afford the neuron the ability to fire multiple spikes after a threshold crossing. Also, when the AP is added, the standard hard reset of the neuron's potential is removed so as to allow bursting (as will be shown in the equations) but since the AP is always inhibitory it can also mutate to prevent the neuron from spiking after the initial spike. The concept of the spiking AP has been formalized in [33], and later, its contribution to bursting has been investigated in [34]. The implementation of the spiking AP here is minimalistic, while achieving the desired outcome of bursting. The equations for

the modified LIF in differential form are represented by Eq 1.

$$
\begin{aligned}
\frac{\mathrm{d}u_{i,j}^L(t)}{\mathrm{d}t} &= \frac{-u_{i,j}^L(t)}{\tau_{i,j}^{syn}} + W_{i,j}\sum_{n=1}^{N}\delta\left(t - t_{i,j,n}^{L-1} + D_{i,j}^{L-1}\right) \\[2mm]
v_i^L(t) &= \sum_{j=1}^{K} u_{i,j}^L(t) + A_{i,j}^L(t) \\[2mm]
S_i^L(t) &= \mathcal{H}\left(v_i^L(t) - v_{th}\right) \\[2mm]
\frac{\mathrm{d}A_{i,j}^L(t)}{\mathrm{d}t} &= \frac{-A_{i,j}^L(t)}{\tau_{i,j}^{ap}} + \beta_{i,j}S_i^L(t)
\end{aligned}
\tag{1}
$$

These equations describe multiple pre-synaptic neurons in layer $L - 1$, indexed by $j$, converging on a single postsynaptic neuron in layer $L$, indexed by $i$. In these equations, $v(t)_i$ is the somatic voltage, $u_{i,j}$ is the synapto-dendritic voltage, which incorporates the contributions of the synaptic kernels and the dendritic filtering. Thus, Eq 1 summarizes the postsynaptic potential contribution from neurons $j$ to neuron $i$. $A_{i,j}$ is the per-dendrite AP feedback. $\tau_{i,j}^{syn}$, $W_{i,j}$ and $D_{i,j}$ are the synaptic time constant, synaptic weight and axonal delay respectively between neurons $j$ and $i$. $t_{i,j,\,n}$ is the time, indexed by $n$, of a spike arriving from neuron $j$ to neuron $i$. This time belongs to the set of all spike times in a train between two neurons $\{t_1, t_2, t_3, \ldots, t_n, \ldots, t_N\}$. $\delta$ is the Dirac delta function. $N$ and $K$, are the total number of incoming spikes and presynaptic neurons (previous layer) respectively. $S(t)$ is the output spike train of the post synaptic neuron, $\mathcal{H}$ is the Heaviside function and $v_{th}$ is the threshold voltage. $\tau_{i,j}^{ap}$ is the AP time constant and $\beta_{i,j}$ is the AP scale between neurons $j$ and $i$. The parameters $\tau^{ap}$ and $v_{th}$ are constant and have the values 4 ms and 1.1 mV respectively. Regarding the reset operation, if no afterpotential is involved, it is a hard reset, where $v(t)$ is forcibly pulled down to zero. By contrast, if the afterpotential is involved, no hard reset is required as the afterpotential itself can pull the somatic voltage even below zero (or realistically more negative i.e hyperpolarization).

The above differential equations are discretized and presented analytically in Eq 2 following the convention used by [12, 35], where $\Delta t$, the time step, is set to 1 ms. A simple visualization of these dynamics is provided by Fig 1B, illustrating the contribution of three pre-synaptic neurons to the somatic voltage of a single postsynaptic neuron while highlighting some of the mutable parameters.

$$
\begin{aligned}
I_{i,j}^L(t) &= W_{i,j}\sum_{n=1}^{N}\delta(t - t_{i,j,n}^{L-1} + D_{i,j}^{L-1}) \\[2mm]
v_i^L(t+1) &= \sum_{j=1}^{K}\exp\left\{\frac{-1}{\tau_{i,j}^{syn}}\right\}u_{i,j}^L(t) + I_{i,j}^L(t) + A_{i,j}^L(t) \\[2mm]
S_i^L(t+1) &= \mathcal{H}(v_i^L(t+1) - v_{th}) \\[2mm]
A_{i,j}^L(t+1) &= \exp\left\{\frac{-1}{\tau_{i,j}^{ap}}\right\}A_{i,j}^L(t) + \beta_{i,j}S_i^L(t+1)
\end{aligned}
\tag{2}
$$

### The evolutionary algorithm

The evolutionary algorithm we used is an example of a 'Natural Evolution Strategy' [36] and is visualized in Fig 1C. The evolutionary procedure starts with the random initialization of a population of (hundreds of thousands) solutions. These solutions are then evaluated for their performance. The loss function used in their evaluation depends on the type of output—if the output is a spike count, mean square error (MSE) is used, whereas if the output is a spike train, an exponential decay kernel, with a 5 $ms$ time constant, is applied to both the output and target [37] before calculating the MSE. The solutions are then sorted according to their loss, and the top performing solutions (elites) are chosen as parents for the next generation. The number of elites chosen are in the thousands. Following their selection, the elites are mutated by adding randomized vectors to their parameters. These vectors have normally distributed values with zero mean and unit variance ($\sim N(0, 1)$), but multiplied by a scaling factor which is referred to as the mutation rate (MR). The MR is analogous to the learning rate in backpropagation-based learning, a high MR would greatly differentiate a child solution from its corresponding parent. Finally, it should be noted that the values any parameter can take during evolution are restricted to fall within a specified range as [$Value_{min}$, $Value_{max}$]. This constraint is a form of regularization and will be referred to hereafter as the clipping range.

Note, we are not committed to this specific evolutionary algorithm, and indeed, we could have used other optimization algorithms such as surrogate gradient decent [38] to explore the benefits of modifying temporal parameters. However, we decided to work with a standard evolutionary algorithm for two reasons: a) evolutionary algorithms are easier to use with spiking networks given that no gradients need to be computed, and b) the diversity of neurons in the brain is the product of evolution, so it seemed more elegant to use an evolutionary algorithm. Regardless, we expect that similar outcomes would be obtained with alternative methods if applicable.

## Results

### Adapting delays but not weights is necessary to solve a set of semi-temporal logic problems

We first evolved spiking neural networks to implement boolean operators like AND, OR and XNOR encoded in semi-temporal form, that is, when inputs were encoded as temporal sequences and outputs decoded by spike count. The temporal nature of the inputs are illustrated in Fig 2B, where the first entry on the $x$-axis represents the input encoding 001 (NO) and 011 (YES), which means that the spike trains might take the form (......|......) for NO, and (.......||.......) for YES. By contrast, the output is a spike count, with the first entry on the $x$-axis either 0 (YES) or 1 (NO). Different combinations of spike trains and spike counts are applied to each logic problem in five adaptation conditions ($W$, $W\tau_c$, $D\tau_c$, $WD$, and $WD\tau_c$).

We find that evolving weights alone does not result in solutions to all problems as shown in (Fig 2A and 2B leftmost grid). These subfigures show the number of generations needed to reach a perfect solution (zero loss) for a given combination of i) co-mutated parameters, ii) input-output encoding, iii) logic problem type, and iv) weight clipping range. The weight clipping range is a hyperparameter that is used during evolution to restrict the values a parameter, in this case weights, can take. We use subthreshold ([-1, 1]) mV and suprathreshold ([-2, 2]) mV clipping ranges as they exemplify networks that are restricted to values inclusive or exclusive of the threshold value of 1.1 mV. In this figure, each grid pattern represents an average of five trials and each trial involves populations with approximately two million solutions. The number of generations was employed as an indirect measure of the ease of finding a perfect

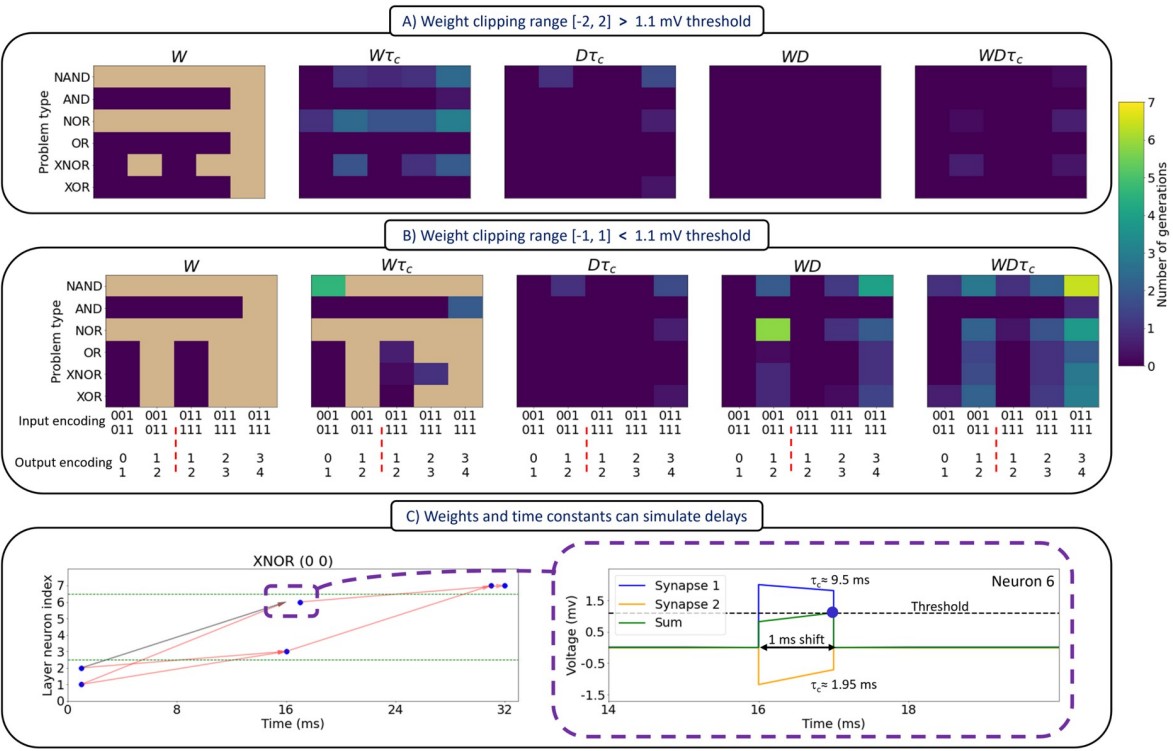

**Fig 2. Effect of the input-output encoding and the co-mutated parameters on the search speed and availability of solutions for various logic problems.** This effect is conveyed through the number of generations needed to find a solution, where the colour 'tan' means no solutions found. This is performed for (A) [-2, 2] mV and (B) [-1, 1] mV weight clipping range during evolution. (C) $W\tau_c$ only solutions can simulate delays. Left, the spiking plot for a sample XNOR problem and right, an enlarged view for the voltage traces with their sum. Abbreviations code, $W$: weights, $\tau_c$: time constants, $D$: delays. For more details, see Table A in S1 Text and the related text.

solution. Thus, from an evolutionary perspective, fewer generations suggests that a given set of neuronal parameters is more favorable to co-mutate to increase the chances of survival of an individual. A trial involves running the evolutionary algorithm for twenty generations.

Weights are atemporal parameters, and when faced with a temporal input (a spike train), we observe that weight-only mutated networks have the lowest performance. Clearly the most important temporal parameters are delays, as all networks that include delays (columns $D\tau_c$, $WD$, and $WD\tau_c$) solve all problems. Delays allow neurons to modify spike arrival times on postsynaptic targets, and when combined with other parameters, greatly enhance the ability of the networks to successfully map input spike trains to spike counts following a diverse set of input-output mappings. Of course, this does not mean that weight-based solutions do not exist. Rather, it shows that in these small feedforward networks, adapting temporal parameters facilitates searching the search space. In addition, we found that the input-output encoding scheme has a significant effect on these results, with the larger the number of spikes in the output, the lower the performance. One possible reason for this is that it gets harder to optimally distribute presyanptic activity to unique spike timings in the output as required by higher output spike counts.

## Weights and time constants can simulate delays

The importance of delays may help explain how networks which only adapt weights and time constants can solve all the problems tested here, because weights and time constants in

combination can simulate delays, as shown in Fig 2C. This result is achieved through the integration of postsynaptic potentials of different time constants. The left image in Fig 2C is a spiking raster plot, which shows the spike arrival times (arrow heads) and spiking times (blue dots) for each neuron in the network (see Fig 1A for the neuron numbers), including the input neurons, namely 1 and 2. The purple-dashed rectangle emphasizes a case where neuron 6 spikes one millisecond later after the spikes from neurons 1 and 2 arrive. This process is expanded on in the right image of Fig 2C, where the voltage traces of each synapse is shown beside their sum. In this example, the sum of a slow excitation and fast inhibition is a rising potential that fires 1 ms later after the arrival of the presynaptic spikes. While inhibition is typically slower than excitation, fast inhibition has been observed in several systems (for example, [39]).

## Adapting delays and time constants can solve all the semi-temporal logic problems

Delays and time constants can solve all logic problems as shown in the middle of Fig 2A and 2B. Example solutions to the XOR problem in the form of the spiking plots are given in Fig 3C. One interpretation of this is that time constants share some functionality with weights given that weights in combination with delays are well suited to solve all logic problems, as shown in the second image from the right in Fig 2. Specifically, short time constants can emulate weight-based self-inhibition by preventing two successive spikes from eliciting a meaningful response. Combined with the ability of delays to temporally separate spikes from the presynaptic neurons, this can lead to solutions which do not rely upon weight adaptation.

In additional simulations (not included here), networks with somatic time constants failed to converge to a solution under the same simulation conditions. It is not immediately clear why this might be the case. However, a possible reason might be that input units need to contribute contrastive time constants to each hidden unit, and indeed, this is what we found with our solutions as shown in the top middle left of Fig 3C. By contrast, if the time constants were somatic, each column would necessarily have the same value, and this may prevent a solution. The contrastiveness we observed might also help to explain the bimodal nature of the distributions shown on the bottom left of Fig 3C. However, it must be noted that the clipping constraint during evolution is a factor that contributes to the bimodal distributions of the delays and time constants. In this case, further analysis is needed to disentangle these two factors, for example, by applying different constraints during evolution.

Finally, another source of evidence for the shared computational roles of weights and time constants is shown in Fig 3D. It can be seen that both distributions show a similar pattern of modulation to the output encoding. Increasing the number of spikes in the output shifts the parameter distributions towards more excitatory connections and also longer time constants.

## The weight clipping range determines the mode of computation

The weight clipping range, as shown in Fig 3B, mainly affects two properties of the solutions: the mode of computation and the EI ratio. The mode of computation refers to the methods by which presynaptic neurons elicit a response in the postsynaptic neuron. When the clipping range is below threshold, presynaptic neurons need to cooperate to induce a spike in the postsynaptic neuron. This mode can be referred to as 'feature-integration', and is exemplified by the top two spiking plots of Fig 3B. Alternatively, when the clipping range is above threshold, a single spike from the pre-synaptic neuron may be sufficient to elicit a response in the postsynaptic neuron. This mode can be referenced to as 'feature-selection', and is exemplified by the bottom two spiking plots of Fig 3B. This dual nature of computation also explains the difference in the EI ratio between the weight clipping ranges examined. For the feature-integration

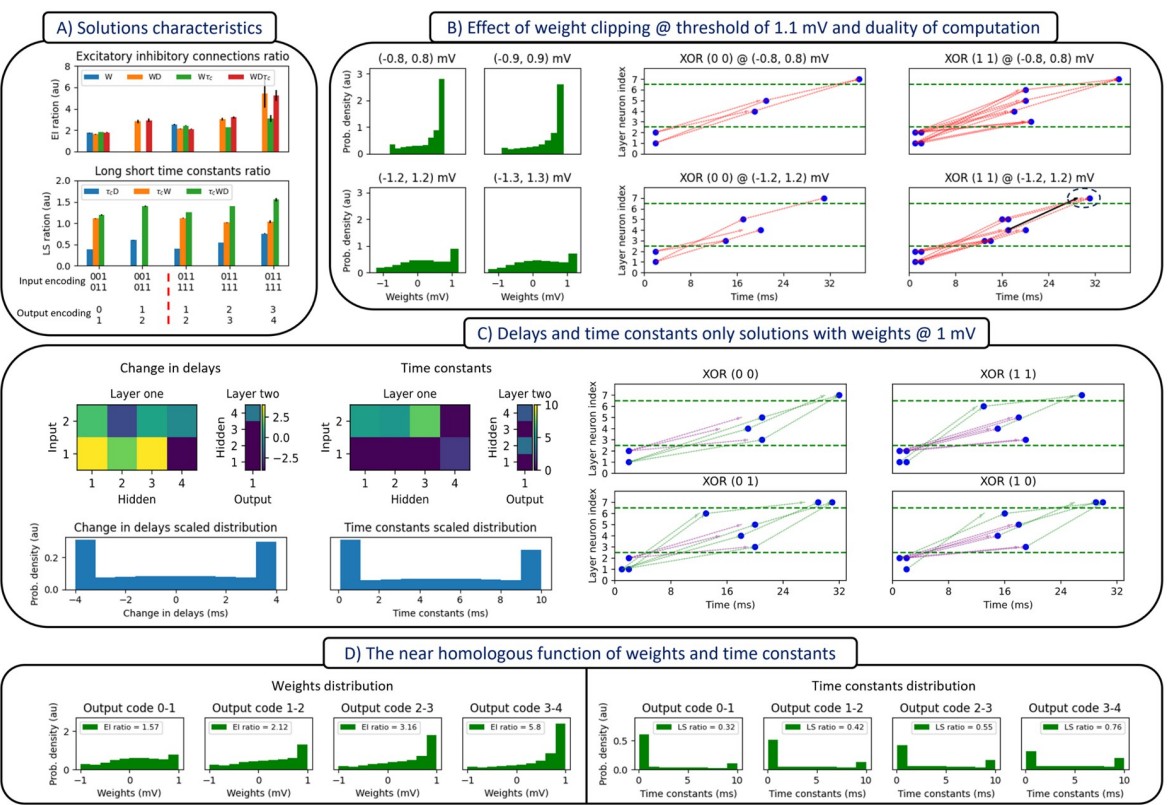

**Fig 3. Properties of semi-temporal logic problems.** (A) Solutions characteristics; demonstrating the impact of the co-mutated parameters and the input-output encoding on the ratios of excitatory vs. inhibitory connections and long vs. short time constants. (B) The weight clipping range dictates the modality of computation and excitatory/inhibitory ratio. Left, distribution of weights in the network at various weight clipping ranges. Right, spiking raster plots illustrating the network behavior when the weight clipping range is below (top row) and above threshold (bottom row). (C) Delays and time constants alone can solve all logic problems with constant weights (1 mV). The heatmaps show the values of change in delays and time constants for the particular XOR problem in the accompanying spiking raster plots (right), while the histograms show the distributions for both parameters aggregated across all logic problems. Regarding delays, we mutate/adapt the change in delays and add it to a default value to acquire the total axonal delays (D) Weights and time constant distributions as a function of the output code. For more details, see Tables B, C and D in S1 Text and the related text.

mode, more cooperation between excitatory spikes is needed to produce the desired number of spikes in the output. In contrast, this is not needed for the feature-selection mode, where inhibition is favoured more so as to dampen excessive excitation, as shown by the dashed ellipse in the bottom right most raster plot of Fig 3B. However, these computational modes are not exclusive, as the suprathreshold clipping case ([-2, 2] mV) was found to exhibit both modes.

In addition, as shown in Fig 3A, both the Excitation/Inhibition (EI) weight ratio and the long/short time constant ratio also depend on which parameters are co-adapted. This interdependence is also a function of the input-output encoding as seen by the different EI ratios and long/short term constant ratios across various parameter combinations. For the EI weight ratio, the most important factor is the increase in the number of output spikes, as reflected in the increasing EI ratio with a larger number of output spikes.

## Multiple parameter distributions support equivalent solutions

A generalization of the fact that weight clipping ranges can lead to qualitatively different solutions to the same logic problem would be the following claim: multiple interacting neural

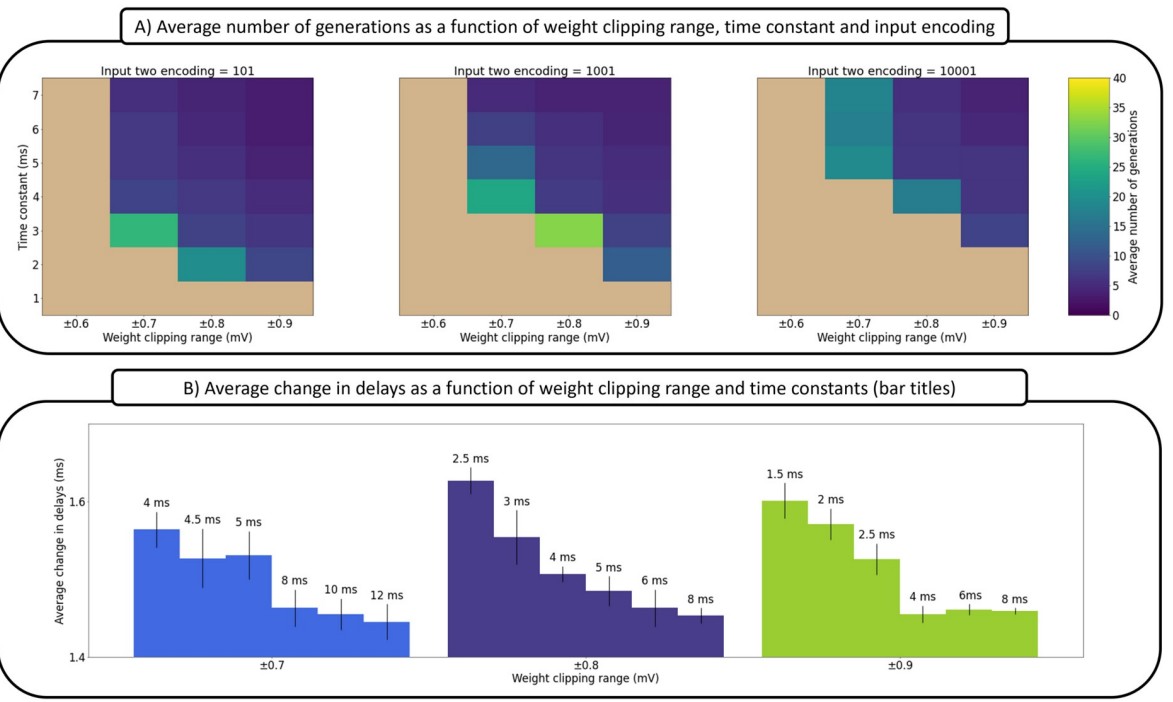

**Fig 4. The relationship between time constants, input encoding and weight clipping range in weights-delays mutated solutions for the `XOR` problem.** (A) This interplay is captured through the average number of generations needed to reach a zero loss (perfect) solution, where each grid cell is the average of five solutions. (B) Average change in delays as a function of the weight clipping range and time constants. Each bar is the average of several populations of solutions that solve the `XOR`, `XNOR`, `OR`, and `AND` logic problems. For more details, see Tables E and F in S1 Text and the related text.

mechanisms can solve the logic problems in different ways as long as there is enough flexibility in their dynamic range. This is exemplified by the results shown in Fig 4A, where we show that various ranges of time constants, weights and input encodings can solve the `XOR` problem. For the input encodings, one input was fixed (`01`) and the other, "input two" varied in duration (i.e., `101`, `1001`, `10001`). In these simulations, weights and delays are adapted while time constants and weight clipping ranges are treated as hyperparameters of evolution, with time constants and weight clipping ranges systematically varied across conditions.

It is evident from the results shown in Fig 4A that the search speed is negatively impacted by the increase in the length of the input encoding, and the decrease in the weight clipping range and time constants. For the input encoding case, longer time constants are needed to communicate more distant spikes. However, since time constants are fixed as a hyperparameter, there is an evolutionary pressure driving delays towards larger values for longer input encodings. Thus, we observe a decrease in performance with longer input encodings because the delays search space increases due to the need for longer delays. This outcome might also help explain the decrease in performance with shorter time constants, as again longer delays are necessary.

In addition, the results highlight the complimentary roles of time constants and delays. This can be observed in Fig 4B, where, on average, longer delays are required to complement shorter time constants. This outcome is another manifestation of a shared functionality, similarly to the noted relationship between time constants and weights, here it is between delays and time constants. The time constants in this figure (bar titles) were sampled across a range of values where solutions exist.

## Weights and time constants are negatively correlated at a fixed somatic activity level

In another manifestation of the shared functionality between weights and time constants, we observe that when we hold the somatic activity/potential fixed, the weights and time constants have an inverse relationship. That is, if we decrease the weights values, and at the same time want to keep the somatic potential unchanged, we need to increase the time constants. This can be observed in the leftmost image of Fig 4A, when one moves diagonally across the grid. An indirect way to emphasize this trend is by building a relationship between the change in time constants and the change in weights at a fixed somatic potential. Somatic potentials are a measure of activity, thus keeping them fixed is analogous to keeping the activity/output pattern fixed. Thus, we fix the somatic potential and ask how much change in time constants is needed to compensate a change in weights. This argument is parameterized by the equations in Eqs 3–5.

$$v_s(t) = \sum_{i=1}^{M} w^i \sum_{n=1}^{N} \delta(t - t_n^i) e^{\frac{-t+t_n^i}{\tau_c^i}} \tag{3}$$

$$v_s = w\left(1 + e^{\frac{-T}{\tau_c}}\right) \tag{4}$$

$$\frac{d\tau_c}{dw} = -\frac{\frac{\partial v_s}{\partial w}}{\frac{\partial v_s}{\partial \tau_c}} = -\left(\frac{\tau_c^2}{wT}\left(1 + e^{\frac{T}{\tau_c}}\right)\right) \tag{5}$$

Eq 3 is a general form of a single postsynaptic neuron receiving spike trains from $M$ presynaptic neurons, where each spike train has $N$ spikes, and where $v_s$ is the somatic voltage. The second equation is a simplification under the assumption that there is only one presynaptic neuron emitting two spikes separated by time $T$.

As seen from Equation Eq 5, a change in weights can be compensated by an opposite but nonlinear change in the time constants. Although the nonlinearity is not readily evident in the plots, the relationship is still highly non proportional as the boundaries of weights change by 0.1 mV while time constants change by 1 ms, as seen on the $y$-axis. Thus, at a fixed object encoding length, smaller weight clipping ranges demand longer time constants for successful mappings. However, if the time constants are fixed, then delays need to take larger values which, in turn, increase the delays search space (as discussed before in the former Subsection). In short, when the weight clipping range is small and the time constants are short, delays need to evolve larger values, which makes it harder to find solutions quickly. Finally, it should be noted that, although the weights search space shrinks from decreasing the clipping range, it seems that it is overshadowed by the larger increase in the delays search space, negating any decrease in search speed.

## Adapting temporal parameters reduces the impact of noise in inputs and weights

An important property of neural networks is their robustness to noise both intrinsic (associated with the units themselves) and extrinsic (from their inputs). Given the ubiquity of neuronal noise, it is expected that evolution would select for mutations that are robust to various forms of noise.

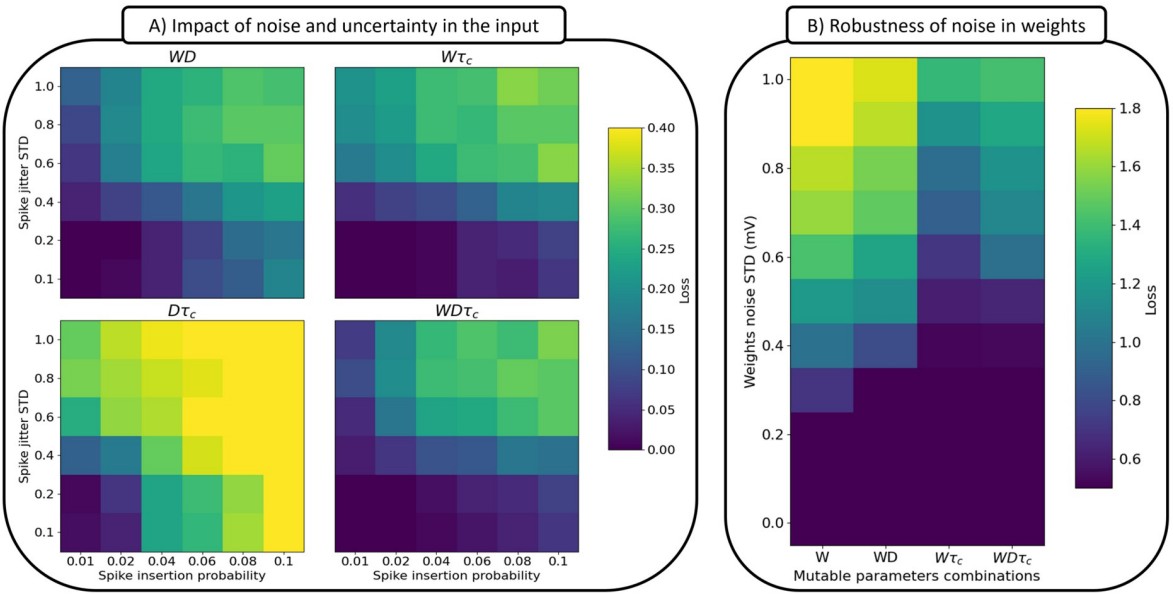

**Fig 5. The effect of noise and uncertainty in inputs and weights.** (A) Impact of additive noise and spike jitter in the input, quantified by the minimum loss achieved within 100 generations. This is shown for weights-delays, weights-time constants, delays-time constants and weights-delays-time constants mutated solutions. (B) Robustness of solutions trained to minimize noise in weights quantified by the minimum loss in 1000 generations, with each loss averaged over 100 trials. This is shown for weights only, weights-delays, weights-time constants and weights-delays-time constants mutated solutions. For more details, see Tables G and H in S1 Text and the related text.

To examine the impact of input noise, we: i) added random spikes that can appear anywhere in the spike train, systematically varying the spike insertion probability, and ii) added temporal jitter to the spike times of the original spike train, systematically varying the spike jitter standard deviation. The jitter takes the form of a Gaussian distribution centred on the original temporal location with a variance parameterised by $\sigma^2$. The results of these manipulations are shown in Fig 5A for different combinations of adaptable parameters, namely; weights-delays, weights-time constants, delays-time constants, and weights-delays-time constants.

To examine the effects of unit (neural) noise, we focused on the weights. Specifically, at each generation, each child solution is copied a hundred times, and for each solution, a different random vector $\sim N(0, \sigma^2)$ is added. This is followed by a loss calculation for each noisy solution, which is then averaged, to give the loss of the initial child solution. Thus, a child solution that minimizes this loss ought to be more resistant to weight perturbations in its life time. The results of these manipulations are shown in Fig 5B, when networks where characterized by adaptable weights-only, weights-delays, weights-time constants, and weights-delays-time constants.

From Fig 5A, it is clear that networks with adaptable delays and time constants (bottom left) were the least robust to input noise, highlighting the primary importance of adaptable weights in generating robustness to input noise. The three networks that included adaptable weights all performed similarly in the context of noise. However, there is some indication that adaptable delays contribute more robustness to spike jitter compared to time constants (when spike insertion probability is low), and similarly, some indication that time constants contribute more to spike insertion robustness (when spike jitter is low). Accordingly, performance is (marginally) best when all parameters are adaptable. Clearly though, the most important factor is whether weights are adaptable or not when faced with both types of noise.

From Fig 5B, it is clear that networks with adaptable weights show some robustness to noise in the weights, and that the addition of delay adaptation further improved robustness. Critically, the addition of time constant adaption played a much larger role in evolving robust networks. The importance of time constants in this context may relate to the overlapping functionality between weights and time constants discussed above (Fig 3D). In this case, the time constants that were not perturbed by noise, and accordingly, the adaptable time constants were able to compensate for the decrease in performance due to weight noise during evolution.

These results may prove important for the design of neuromorphic systems which simulate spiking neuron models. Thus, we can expect at least two sources of noise: i) sensor/input noise and ii) uncertainty in the hardware realization of the network parameters. Mitigating these two types of noise is of paramount importance for a reliable hardware implementation of spiking neural networks. In this regard, we have shown that supporting weight based implementations with adaptable temporal parameters can decrease the impact of both forms of noise.

### Bursting parameter was necessary for fully spatio-temporal tasks

It has been argued before that the brain learns spatio-temporal mappings [28, 29]. Thus, a reasonable next step is to test these temporally adaptive networks on spatio-temporal mappings, which in turn also increase the problem complexity. We achieve these kind of mappings by replacing the spike count output code with a spike train, and in one simulation added adaptive spike Afterpotential (AP) or bursting parameter ($\beta$). In Fig 6A, the number of successful solutions (from five runs) before generation 200 is depicted, and in Fig 6B, the average number of generations needed to reach a perfect solution is shown.

It is evident that as we increase the number of parameters in the neuronal model, it gets progressively easier to evolve perfect solutions. These results are also mirrored with the number of solutions found. Indeed, performance is quite poor in the $W\tau_c$, $D\tau_c$ and $WD$ conditions (conditions in which models performed quite well in solving the semi-temporal logic problems). All three of these adaptive parameters were required in order to do well, with the bursting parameter required to solve all problems with all input and output encoding conditions.

Lastly, the input-output encoding scheme may act as a form of regularization which impacts the distributions of the adapted delay parameters. In other words, and as shown in Fig 6C, the output spike trains can act as a boundary condition for the adapted delays. Systematically changing the temporal position of the output spikes (shown from left to right), leads to a systematic shift in the distributions of changes in delays, where the values start with a heavy bias towards large values (delays increase) then shifts incrementally to more negative values (delays decrease).

## Discussion

Despite the important role that spike timing plays in neural computations and learning, most artificial neural networks (ANNs) ignore spikes, and even spiking neural networks (SNNs) tend to restrict learning to the adaptive modification of atemporal parameters, namely, weights and biases. In a series of simulation studies we highlight the importance of adaptively modifying multiple temporal parameters (time constants, conduction delay, and bursting) in small feed-forward networks in an evolutionary context. Our findings suggest that current network models of the brain are ignoring important dimensions of variation in neurons that may play a key role in how neural systems compute and learn.

In our first simulation we adapted weights, time constants, and conduction delays (but not bursting) parameters and considered semi-temporal logic (XOR, XNOR, OR, NOR, AND, and

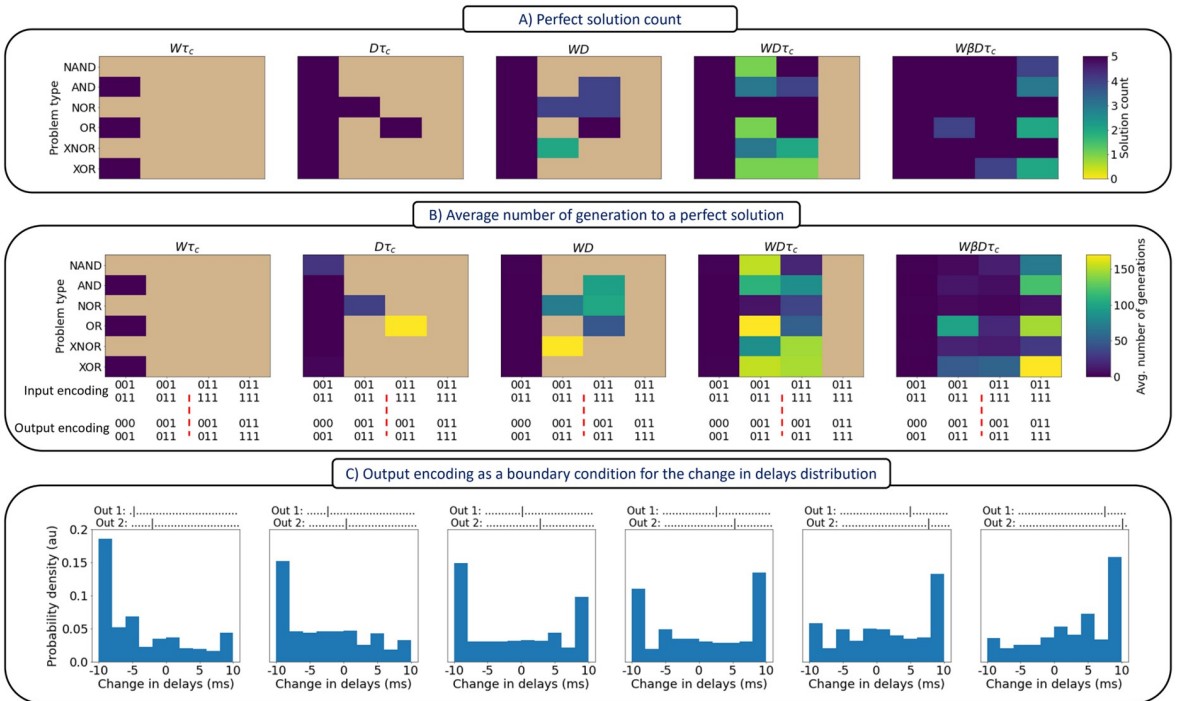

**Fig 6. Effect of the input-output encoding and the co-mutated parameters on the search speed and availability of solutions for various logic spatio-temporal problems.** These relationships are emphasized through (A) the number of perfect solutions found, and (B) the average numbers of generations needed to reach a perfection solution. (C) Manipulating the change in delay distributions through the output spike trains (shown on top of each figure). Abbreviations code, $W$: weights, $\tau_c$: time constants, $D$: delays, $\beta$: spiking afterpotential. For more details, see Tables I and J in S1 Text and the related text. For a bursting example, see Fig A in S1 Text.

NAND) problems, where the input is a spike train and the output is a spike count. In this context, the importance of temporal parameters was evident both in terms of the models' successes in solving a range of problem as well as the number of generations required. Indeed, weight-only adaptation only solved a subset of the problems, whereas all problems could be solved without modifying weights. In addition, the adaptive parameters interacted in unexpected ways, with pairs of parameters combining to simulate the impact of another, or sharing functionality with another. For example, we found that weights and time constants sometimes simulated delays (see Fig 2C). Similarly, weights and time constants are affected by a systematic increase in the number of spikes in the output (see Fig 3D), suggesting that they share functionality. Together, these observations may help to explain why delays and time constants can solve all logic problems.

We have also shown that multiple evolved solutions exist for the same problem and that these solutions can differ qualitatively. For example, the weight clipping range dictates the mode of computation. When individual synapses can be strong enough to drive a neuron to fire following a single spike, we find the network often engages in the feature selection mode by firing in response to a single spike from a single input. By contrast, when individual synapses are weaker and cannot drive units beyond threshold, the network necessarily engages in feature integration by combining multiple spikes from multiple inputs before firing. Typically, weights are not clipped in SNNs, but real neurons are rarely (if ever) driven by a single spike, and accordingly, the feature integration mode is more biologically plausible. Still, there are cases in which bursts from a single presynaptic neuron can drive a postsynaptic neuron. For

example, in the domains of motor control, [40] reported that a train of action potentials in a single pyramidal cell of rat primary motor cortex can cause whisker movement, and in the case of sensory systems, rats could perceive the microstimulation of somatosensory (barrel) cortex that produced a train of action potentials in a single neuron. Clearly, the strengths of synapses vary across systems, and our findings suggest this will have important impacts on the nature of the underlying computations. More generally, varying the hyperparameters of evolution (weight clipping range, input encoding, etc.) can lead to equivalent solutions for the same problem, with different distributions of adapted parameters observed across conditions.

Importantly, different forms of adaption were important for coping with different types of noise. In the case of input noise (in the form of adding or displacing spikes to the input), we found adapting weights was most important, albeit a slight boost was gained by adapting all the morphological parameters. In contrast, when introducing noise in the weights, co-adapting weights with other temporal parameters lessens the impact of the uncertainty in the weights values, with time constants being particularly important. For software implementations of neural networks, it is mainly the input noise that is an issue, while for hardware implementations, it can be both the input noise and parameter noise as reflected in the uncertainty in the parameter values. Accordingly, our results suggest that adapting temporal parameters may be important in the development of neuromorphic systems which are robust to noise and manufacturing inconsistencies.

Finally, when evolving networks on spatio-temporal tasks (mapping spike trains to spike trains), the importance of adapting multiple spatio-temporal parameters is even more apparent. Weights and time constants alone, or weights and delays alone could not solve the task as reliably as when all three parameters are co-mutated. This enforces the need to incorporate multiple relevant forms of temporal adaptation when tackling tasks of this sort. This claim is further supported by the fact that adding a third form of temporal adaption, namely bursting, greatly enhances the performance in mapping spatio-temporal spike patterns.

Of course, these findings have been obtained in highly idealized and simplified conditions, with tiny networks composed of between 7 and 9 units adapted to solve boolean logic problems with a standard natural evolution algorithm. These are quite different conditions compared to the environment faced by simple organisms billions of years ago. Nevertheless, we take our findings as a proof of principle that adapting temporal parameters can be advantageous in an evolutionary context, and that these observations may have important implications for both neuroscience and modelling. With regards to neuroscience, our findings may help explain why there are a wide variety of neuron types that vary in their morphological forms in ways that dramatically modify their processing of time. A previous study has shown that variability in membrane and synaptic time constants is relevant when solving tasks with temporal structure (e.g., [12]). Our studies extend these results to time constants, conduction delays, and bursting parameters in an evolutionary context. The evolutionary context is important, as different neuron types are the product of evolution not learning. Of course, in-life learning may not be restricted to changes in synaptic strengths, and indeed, there is good empirical evidence for learning on the basis of time-based parameters [14].

With regards to modelling, our findings are relevant to both the SNN and ANN communities. First, our findings suggest that the current focus on selectively learning or adapting weights in SNNs is too restrictive, and that adding various temporally based parameters associated with spiking units may be important. This includes a bursting parameter that played a crucial role in feedforward mappings of whole spike trains. As far as we are aware, we are the first to map input-output spike trains in a feedforward manner in spiking networks. This may require using many more neurons in SNNs that adapt only weights and biases. Critically, adapting multiple parameters may be important not only for modelling the brain, but also for

improving model robustness to noise. This latter observation may be particularly important if SNNs are to be simulated on neuromorphic systems where various types of noise will be present.

With regards to ANNs, our findings challenge a key assumption underpinning research comparing ANNs to brains. The current excitement regarding ANN-brain alignment is based, in large part, on the ability of ANNs to solve complex real word tasks (such as identifying objects and engaging in dialogue) and the ability of ANNs to predict brain activity better than alternative models, both in the domain of vision and language [41]. Based on these successes, many researchers have concluded that rate coding is a good abstraction for spikes, with the dimension of time reduced to time steps, and the only relevant variation between units being their weights and biases. For example, in describing "The neuroconnectionist research programme", [42] write:

> "ANNs strike the right balance by providing a level of abstraction much closer to biology but abstract enough to model behaviour: they can be trained to perform high-level cognitive tasks, while they simultaneously exhibit biological links in terms of their computational structure and in terms of predicting neural data across various levels—from firing rates of single cells, to population codes and on to behaviour."

However, our results suggest that adaptation in time-based parameters is not only important in simple tasks, but that quantitatively and qualitatively different types of solutions are obtained under different conditions. This includes changing the distributions of the weights themselves when other adaptive parameters are in play. This raises questions as to whether ignoring spikes and time-based adaptive parameters is the best level of abstraction when building models of brains, and whether the solutions obtained with ANNs are brain-like given the different types of solutions we observed across conditions. Consistent with these concerns, there are some limitations with the evidence taken to support good ANN-brain alignment, including problems with drawing conclusions from good brain predictivity scores [43, 44], and the widespread failure of ANNs to account for empirical findings from psychology [45]. The brain has a language and its syllables are spikes. Our findings suggest that spikes and plasticity of time-based parameters should play a more important role in the current research programs modelling the brain.

## Supporting information

**S1 Text. Supplementary Tables and Supplementary Figures.**
(PDF)

## Author Contributions

**Conceptualization:** Karim G. Habashy, Benjamin D. Evans, Dan F. M. Goodman, Jeffrey S. Bowers.

**Data curation:** Karim G. Habashy.

**Formal analysis:** Karim G. Habashy.

**Funding acquisition:** Benjamin D. Evans, Dan F. M. Goodman, Jeffrey S. Bowers.

**Investigation:** Karim G. Habashy.

**Methodology:** Karim G. Habashy, Jeffrey S. Bowers.

**Project administration:** Benjamin D. Evans, Dan F. M. Goodman, Jeffrey S. Bowers.

**Resources:** Jeffrey S. Bowers.

**Software:** Karim G. Habashy.

**Supervision:** Benjamin D. Evans, Dan F. M. Goodman, Jeffrey S. Bowers.

**Validation:** Karim G. Habashy, Benjamin D. Evans, Dan F. M. Goodman, Jeffrey S. Bowers.

**Visualization:** Karim G. Habashy.

**Writing – original draft:** Karim G. Habashy, Benjamin D. Evans, Jeffrey S. Bowers.

**Writing – review & editing:** Karim G. Habashy, Benjamin D. Evans, Dan F. M. Goodman, Jeffrey S. Bowers.

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
