## [Decision Letter · Decision Letter 0]

19 Aug 2024

Dear Dr. Habashy,

Thank you very much for submitting your manuscript "Adapting to time: why nature evolved a diverse set of neurons" for consideration at PLOS Computational Biology. As with all papers reviewed by the journal, your manuscript was reviewed by members of the editorial board and by several independent reviewers. The reviewers appreciated the attention to an important topic. Based on the reviews, we are likely to accept this manuscript for publication, providing that you modify the manuscript according to the review recommendation, though we cannot, of course, guarantee acceptance of the manuscript at this time. 

As with all papers reviewed by the journal, your manuscript was reviewed by members of the editorial board and by several independent reviewers. In light of the reviews (below this email), we would like to invite the resubmission of a revised version that takes into account the reviewers' comments.

As you will see, both reviewers (and I) found much merit in your paper. However, some valid concerns were raised. In particular, reviewer 1 found the technical aspects of the paper difficult to evaluate, and would have liked some more detail in that regard. Please do assure the code linked in the paper is available before making your resubmission. I think that addressing these concerns will make the paper much stronger and its potential impact much greater.

We cannot make any decision about publication until we have seen the revised manuscript and your response to the reviewers' comments. Your revised manuscript is also likely to be sent to reviewers for further evaluation.

Sincerely,

Alex Doumas

Academic Editor

PLOS Computational Biology

Andrea E. Martin

Section Editor

PLOS Computational Biology

Reviewer's Responses to Questions

**Comments to the Authors:**

Reviewer #1: See attachment.

Reviewer #2: This is an interesting and well-written paper that examines the effect in spiking neural networks of allowing training to vary timing parameters in addition to the standard variation of synaptic weights. This is of current interest, as there is increasing focus on the importance of spike timing in natural neural processing, and there is emerging evidence of ability of neural plasticity to shape time-dependent parameters.

The authors present clear mathematical formulations on which the experiments are run, as well as simple, clear but still challenging problems on which the networks are trained.

I commend the authors on a stimulating introduction and discussion.

The paper is generally very clear.

Minor points:

1. The change in terminology from Greek characters (L116) to slightly different Roman nomenclature (L149) should be flagged more clearly.

2. Fig. 3B left-hand panels would be clearer if the labels (e.g. (-1.2, 1.2) mV ) were inside the associated box

3. Fig 6. These are complex findings presented in a fairly sketchy manner. It would be interesting to see more raster style plots for the outputs of the bursts to see what variability there is within the family of 011 outputs (for instance), and whether this tells us anything useful? Related to the abbreviated presentation, it is not clear to me in Fig. 6c why the delays become shorter when the output spikes are produced later.

**Have the authors made all data and (if applicable) computational code underlying the findings in their manuscript fully available?**

Reviewer #1: **No: **Paper states that the code is deposited on GitHub, but the URL returns a 404. Is the repository private?

Reviewer #2: **No: **The author data availability statement says "The data in this manuscript is simulated using the code Accessible at:

" ext-link-type="uri" xlink:type="simple">https://github.com/biocomplab/NeuroMorph.git"

This link brings me to an empty page. The source should also be listed explicitly in the Methods section.

PLOS authors have the option to publish the peer review history of their article (what does this mean?). If published, this will include your full peer review and any attached files.

Reviewer #1: No

Reviewer #2: No

Figure Files:

Data Requirements:

Reproducibility:

To enhance the reproducibility of your results, we recommend that you deposit your laboratory protocols in protocols.io, where a protocol can be assigned its own identifier (DOI) such that it can be cited independently in the future. Additionally, PLOS ONE offers an option to publish peer-reviewed clinical study protocols. Read more information on sharing protocols at https://plos.org/protocols?utm_medium=editorial-emailutm_source=authorlettersutm_campaign=protocols

References:

---

## [Decision Letter · Decision Letter 1]

25 Nov 2024

Dear Dr. Habashy,

We are pleased to inform you that your manuscript 'Adapting to time: why nature may have evolved a diverse set of neurons' has been provisionally accepted for publication in PLOS Computational Biology.

Best regards,

Alex Leonidas Doumas

Academic Editor

PLOS Computational Biology

Andrea E. Martin

Section Editor

PLOS Computational Biology

Feilim Mac Gabhann

Editor-in-Chief

PLOS Computational Biology

Jason Papin

Editor-in-Chief

PLOS Computational Biology

Reviewer's Responses to Questions

**Comments to the Authors:**

Reviewer #2: I am satisfied with the revisions to the manuscript

**Have the authors made all data and (if applicable) computational code underlying the findings in their manuscript fully available?**

Reviewer #2: Yes

PLOS authors have the option to publish the peer review history of their article (what does this mean?). If published, this will include your full peer review and any attached files.

Reviewer #2: **Yes: **Richard M Vickery

---

## [Editor Report · Acceptance letter]

6 Dec 2024

PCOMPBIOL-D-24-01038R1 

Adapting to time: why nature may have evolved a diverse set of neurons

Dear Dr Habashy,

I am pleased to inform you that your manuscript has been formally accepted for publication in PLOS Computational Biology. Your manuscript is now with our production department and you will be notified of the publication date in due course.

With kind regards,

Anita Estes
